# Fast viral dynamics revealed by microsecond time-resolved cryo-EM

Oliver F. Harder [1,2], Sarah V. Barrass [1,2], Marcel Drabbels [1] &
Ulrich J. Lorenz [1] ✉

Observing proteins as they perform their tasks has largely remained elusive, which has left our understanding of protein function fundamentally incomplete. To enable such observations, we have recently proposed a technique that improves the time resolution of cryo-electron microscopy (cryo-EM) to microseconds. Here, we demonstrate that microsecond time-resolved cryo-EM enables observations of fast protein dynamics. We use our approach to elucidate the mechanics of the capsid of cowpea chlorotic mottle virus (CCMV), whose large-amplitude motions play a crucial role in the viral life cycle. We observe that a pH jump causes the extended configuration of the capsid to contract on the microsecond timescale. While this is a concerted process, the motions of the capsid proteins involve different timescales, leading to a curved reaction path. It is difficult to conceive how such a detailed picture of the dynamics could have been obtained with any other method, which highlights the potential of our technique. Crucially, our experiments pave the way for microsecond time-resolved cryo-EM to be applied to a broad range of protein dynamics that previously could not have been observed. This promises to fundamentally advance our understanding of protein function.

Cowpea chlorotic mottle virus is an icosahedrally symmetric plant virus in the *Bromoviridae* family that infects cowpea plants (*Vigna unguiculata*)[1]. As with most viruses, CCMV faces the challenge of safely packaging its genetic material for transport, but then releasing it at the appropriate time to infect the host. As illustrated in Fig. 1, CCMV is thought to achieve this by detecting a change in its chemical environment upon entering the host cell, which causes its capsid to swell, increasing in diameter by about 10%[2]. This extended state, which is unstable and disassembles, releases the viral RNA, thus infecting the host[3]. The capsid swelling is triggered by a decrease in the concentration of divalent ions and a simultaneous increase in pH upon entry into the host cell. This causes calcium ions to vacate their binding sites on the capsid interior, where they are complexed by several negatively charged sidechains[3,4]. Once the calcium ions are no longer present to compensate these negative charges, electrostatic repulsion causes the capsid to expand[3–5]. In the absence of divalent ions, the virus can also be artificially contracted by lowering the pH below 5,

which protonates the negatively charged residues and thus removes their repulsion[6].

Our understanding of how the CCMV capsid functions has remained incomplete due to a lack of direct observations of the fast motions of this intricate nanoscale machine, or even methods that would enable such observations. A comparison of the contracted and expanded virus structures[6] suggests that the mechanics of the capsid motion must involve several large-scale translations and rotations of the capsid proteins. However, it is unclear whether they occur in a concerted or asynchronous fashion[3,5]. Moreover, it is unknown how fast these motions are. The pH induced contraction of another icosahedral virus, Nudaurelia Capensis ω, was observed to be complete after 10 ms, but is thought to be significantly faster[7]. This suggests that the capsid motions of CCMV would be too fast to be captured by traditional time-resolved cryo-EM, which affords only millisecond time resolution[8]. Ultrafast x-ray crystallography[9,10], while sufficiently fast, is likely not suitable either to study these dynamics, since the crystal

[1]Ecole Polytechnique Fédérale de Lausanne (EPFL), Laboratory of Molecular Nanodynamics, CH-1015 Lausanne, Switzerland. [2]These authors contributed equally: Oliver F. Harder, Sarah V. Barrass. ✉e-mail: ulrich.lorenz@epfl.ch

environment would hinder the large-amplitude motions involved[10,11]. The difficulty of observing the fast motions of the CCMV capsid exemplifies the broader challenge of observing proteins as they perform their tasks. This has largely remained elusive, which has left our understanding of protein function fundamentally incomplete[12]. In order to enable such observations, we have recently proposed microsecond time-resolved cryo-EM[13–17], which affords a time resolution of about 5 µs or better[13] and enables near-atomic resolution reconstructions[17]. Here, we demonstrate that microsecond time-resolved cryo-EM enables observations of fast protein dynamics.

## Results

Here, we employ microsecond time-resolved cryo-EM to observe the pH jump induced contraction of CCMV and elucidate its capsid mechanics. The experimental approach is illustrated in Fig. 2a–d. Cryo samples of extended CCMV at pH 7.6 (containing no divalent ions) are

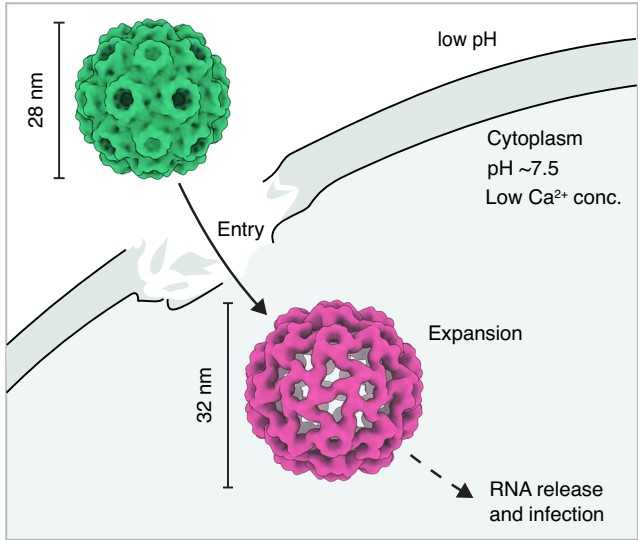

**Fig. 1 | Illustration of the entry of CCMV into the plant cell.** The virus in its contracted state (green) enters the plant cell through damaged sites in the cell wall. Once inside the cytoplasm, the virus experiences a decrease in the concentration of divalent ions and an increase in pH, which causes its capsid to swell. The extended state, which is unstable and disassembles, then releases the viral RNA, thus infecting the host.

prepared in the presence of a photoacid (NPE-caged-proton). The pH of the cryo sample is then lowered to 4.5 by releasing the photoacid through UV irradiation (266 nm). Even though the fully contracted state of CCMV is the most stable at this low pH[18] (Fig. 2e), the matrix of vitreous ice surrounding the particles prevents their contraction, locking them in their extended configuration. However, when we rapidly melt the sample with a laser beam (532 nm, Fig. 2b), the particles begin to contract as soon as the sample is liquid (Fig. 2c). After 30 µs, we then switch off the heating laser, and the sample cools and revitrifies within microseconds, trapping the particles in their partially contracted configurations (Fig. 2d), which we subsequently image.

Figure 3a shows a single-particle reconstruction of CCMV in its extended state (3.9 Å resolution). The sample was plunge-frozen at pH 7.6, after which the pH was lowered to 4.5 by releasing the photoacid. The capsid has a diameter of 32 nm (Materials and Methods), with the disordered RNA in its interior not resolved. The reconstruction is indistinguishable from one obtained without first lowering the pH (Supplementary Fig. 2a). This confirms that the vitreous ice matrix has prevented the particles from contracting, even though the negatively charged residues whose repulsion keeps the capsid inflated are likely protonated due to the high proton conductivity of vitreous ice[19]. In contrast, a sample prepared at pH 5.0 yields the structure of the fully contracted state with a diameter of 28 nm (Fig. 3c). The resolution of 1.6 Å is considerably higher than that of the extended state, which is more flexible and prone to partial disassembly.

When we prepare CCMV in its extended state and lower the pH to 4.5, melting and revitrification of the sample allows the particles to partially contract. A reconstruction from a revitrified sample (Fig. 3b, 8.0 Å resolution) features a particle diameter of 31 nm, which lies in between that of the extended and the contracted configurations. This is also evident in Fig. 3d, which shows a cross section of the three reconstructions overlaid. Since the particles do not contract in samples that are UV irradiated, but do not contain any photoacid (Supplementary Fig. 2b), we conclude that the contraction is induced by the pH jump.

The partially contracted CCMV particles obtained after revitrification feature substantial conformational heterogeneity, which limits the resolution of the reconstruction in Fig. 3b to 8.0 Å. This is confirmed by a variability analysis (cryoSPARC 4.0.1[20]). Figure 3e displays the distribution of the particles in the extended, intermediate, and contracted configurations (21,675 randomly selected particles of each) as a function of the first two variability components. The first component predominantly corresponds to a change in particle diameter, while the second is associated with motions of the capsid

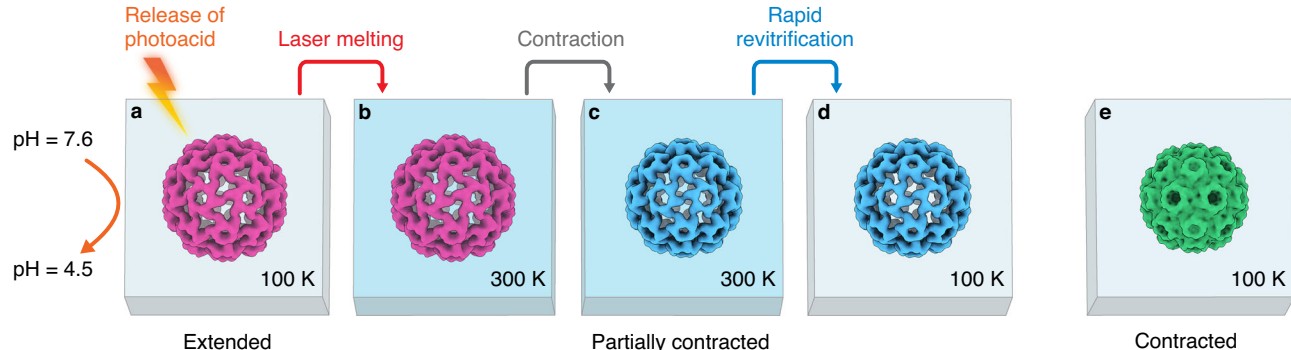

**Fig. 2 | Microsecond time-resolved cryo-EM of the contraction of CCMV— experimental concept. a** The expanded form of CCMV is prepared at pH 7.6 and plunge frozen in the presence of a photoacid (NPE-caged-proton). The pH of the cryo sample is then lowered to 4.5 by releasing the photoacid through UV irradiation. At such a low pH, the contracted state of the capsid (**e**) is more stable. However, contraction cannot occur since the virus is trapped in the vitreous ice. **b** The sample is rapidly melted through irradiation with a laser beam. **c** Once the sample is liquid, the capsid starts to contract. **d** The laser is switched off, which causes the sample to cool and revitrify within microseconds, trapping the virus in partially contracted configurations, which are subsequently imaged. **e** A reconstruction of the fully contracted virus is separately obtained from a sample prepared at pH 5.0.

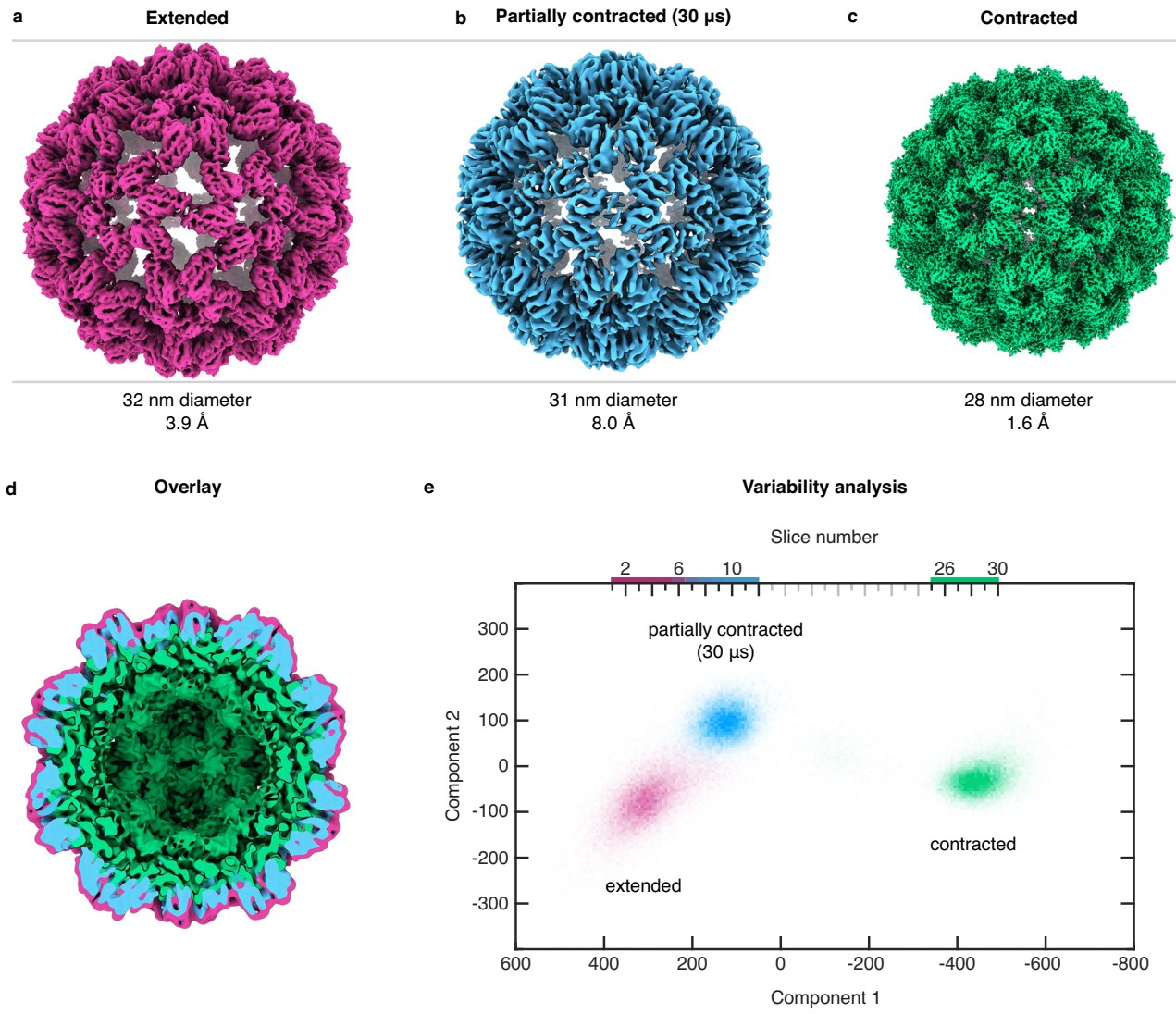

**Fig. 3 | Single-particle reconstructions of different stages of the contraction of CCMV, and variability analysis. a–c** Comparison of the expanded state (plunge frozen at pH 7.6), the partially contracted configuration obtained with a 30 μs laser pulse, and the fully contracted state (prepared at pH 5.0). Before acquiring micrographs of the expanded state, the pH of the cryo sample was lowered to 4.5 by releasing a photoacid through UV irradiation. The particles nevertheless remain expanded because the surrounding matrix of vitreous ice prevents their contraction. **d** Cross section of an overlay of the three reconstructions (filtered to 8 Å), highlighting that the structure obtained after 30 μs of laser irradiation (blue) corresponds to a partially contracted configuration. **e** Variability analysis (cryoSPARC 4.0.1[20]) of the extended, partially contracted, and fully contracted configurations (21,675 randomly selected particles of each). The particle distribution is shown as a function of the first two components, which are predominantly associated with the diameter change of the particle (component 1) and with motions of the capsid proteins (component 2). The particle distribution is divided into 30 slices along the first component, and a reconstruction is obtained for each.

proteins. The three configurations appear as distinct clusters, with the extended and partially contracted ensemble closer to each other and partly overlapping. Interestingly, the variability analysis in Fig. 3e suggests that the reaction path may be curved, indicating that different motions involved in the contraction process occur on different timescales.

An analysis of the translations and rotations of the capsid proteins confirms that contraction involves different timescales. We divide the particle distribution in Fig. 3e into 30 equally spaced slices along the first variability component and perform a reconstruction for each (Supplementary Fig. 4). We then dock atomic models of the extended configuration into slices 1–12 (extended and partially contracted particles) and of the contracted configuration into slices 25–30 (fully contracted particles), from which we determine the motions of the capsid proteins (Supplementary Methods). Figure 4a illustrates these motions, with the icosahedral capsid shown in its extended form. The 180 identical capsid proteins are arranged in 12 pentamers and 20

hexamers. The asymmetric unit (highlighted) contains three subunits A, B, and C (magenta, green, and blue, respectively). Figure 4b displays the particle diameter as a function of the slice number, with the diameter measured along the five-fold symmetry axis (indicated in Fig. 4a). In slices 1–5, which contain extended particles, as well as in slices 25–30, which contain fully contracted particles, the diameter is constant. In contrast, a continuous distribution of diameters is found for the partially contracted particles in slices 8–12, with the particle diameter in slice 12 about halfway between the fully extended and contracted configurations. This wide distribution highlights the conformational heterogeneity of the ensemble obtained after melting and revitrification. Evidently, the particles contract at different speeds.

The contraction of CCMV is accompanied by a simultaneous anticlockwise rotation of the pentamers and hexamers, which are both rotated by about 5 degrees in the fully contracted state. Figure 4c displays the rotation angles as a function of slice number. It reveals that the pentamers rotate about twice as fast as the hexamers,

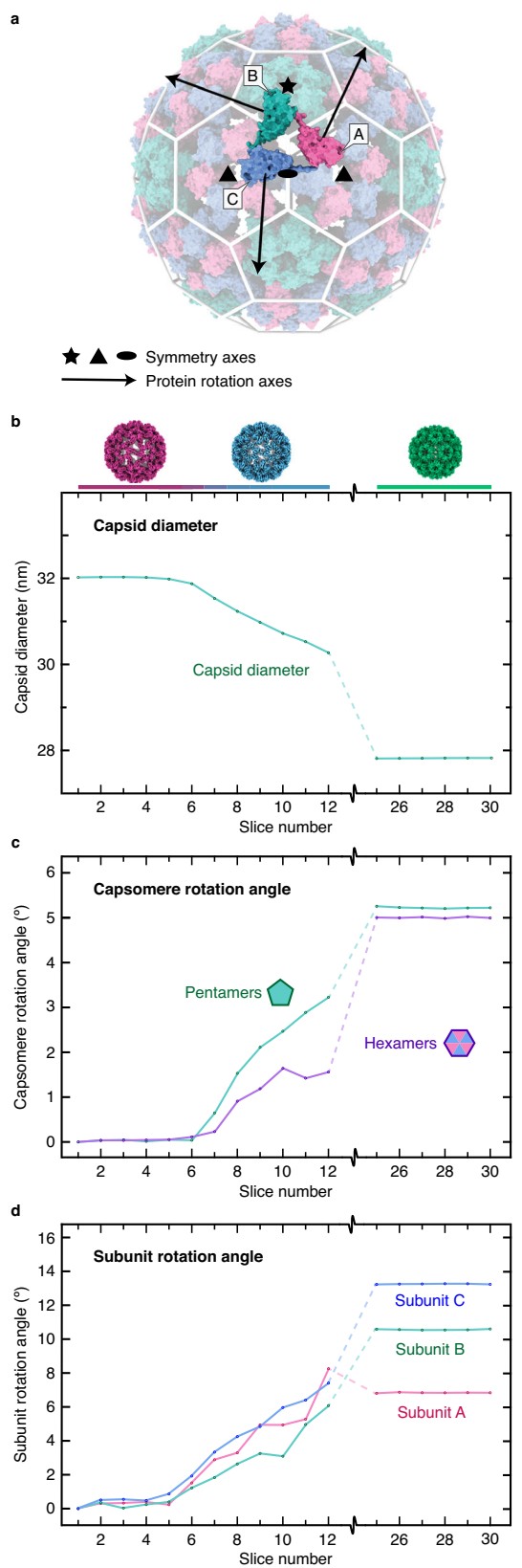

**Fig. 4 | Analysis of the capsid motions involved in the contraction of CCMV.**
**a** Geometry of the icosahedral capsid (extended form). The asymmetric unit (highlighted) contains three protein subunits A, B, and C (magenta, green, and blue, respectively). Five-, three-, and two-fold symmetry axes are indicated with a star, triangle, and ellipse, respectively. Arrows indicate the rotation axes of the capsid proteins. **b**–**d** Analysis of the motions of the capsid proteins. The particle distribution in Fig. 3e is divided into 30 slices along the first component, and reconstructions are obtained for each. Atomic models are then docked into the density, from which the motions of the capsid proteins are extracted for each slice. **b** Particle diameter as a function of slice number. **c**, Rotation angles of the capsid pentamers and hexamers. **d** Angles of the superimposed rotation of the A, B, and C subunits around the rotation axes indicated with arrows in **a**.

However, whereas subunit A has already reached its final rotation angle of ~7 degrees in slice 12, subunits B and C have completed only about half their rotations of ~11 and ~13 degrees, respectively. Clearly, while the different motions of the capsid proteins occur simultaneously, they are associated with different timescales.

Our experiments elucidate the capsid mechanics of the pH jump induced contraction of CCMV. Given the large amplitude of the motions involved, the contraction is surprisingly fast, with some particles completing half the contraction within the time window imposed by the 30 μs laser pulse. The process thus resembles a collapse that is triggered when the electrostatic repulsion is removed that keeps the capsid inflated. While this collapse is a concerted process, the associated translations and rotations of the capsid subunits occur on slightly different timescales, which results in a curved reaction path. Our analysis also reveals a large spread in the speed with which the particles contract. This is an expected result since the contraction occurs in a dissipative medium. For the same reason, conformational heterogeneity will likely be a feature of most protein dynamics than can be observed with microsecond time-resolved cryo-EM. It is therefore advantageous to design experiments such that they start from a homogeneous ensemble. As we have shown here, conformational sorting[21,22] will be crucial to obtain detailed structural information and elucidate reaction paths. We note that CCMV exists in three virions, which each package a different RNA strand[23,24] and will therefore likely contract at slightly different speeds. A further contribution to the observed spread in contraction speeds will likely arise from small variations in the temperature evolution of the revitrified area[13].

While we have previously characterized our technique with the help of proof-of-principle experiments, we here show that microsecond time-resolved cryo-EM can be successfully employed to study fast protein dynamics that occur in vivo. We demonstrate a general approach for triggering such dynamics with the help of photorelease compounds[14]. Instead of uncaging the compound while the sample is liquid, we already do so with the sample still in its vitreous state. This offers the advantage that much larger changes in the chemical environment of the embedded particles can be induced, in particular for caged compounds with small quantum yields. Our experiments confirm that while the particles remain trapped in the matrix of vitreous ice, they cannot react to this stimulus, but will only begin to undergo conformational dynamics once the sample is melted with the laser beam. It should be possible to extend this principle to a wide range of other stimuli that can be applied with photorelease compounds, including caged small molecules, ATP, ions, amino acids, or peptides[25,26]. This suggests that microsecond time-resolved cryo-EM will be broadly applicable and that it has the potential to elucidate the dynamics of a wide variety of proteins that previously were too fast to be observed.

It is difficult to conceive how a similarly detailed view of the fast structural dynamics of the CCMV contraction could have been obtained with any other technique. This is particularly true for methods that do not offer a sufficiently high time resolution and are therefore limited to observing proteins at equilibrium. For example, an

adopting a rotation angle of over 3 degrees in slice 12, while the hexamers reach only about 1.5 degrees. This rotation of the capsomeres is accompanied by a superimposed rotation of the capsid subunits around the axes indicated with black arrows in Fig. 4a. This rotation causes the capsomeres to adopt a domed structure in the contracted state. Figure 4d reveals that subunits A–C rotate with similar speeds.

ordinary conformational analysis of a cryo sample prepared under equilibrium conditions would be unable to access the partially contracted transient configurations that we observe. This highlights the need for fast observations of protein dynamics under out-of-equilibrium conditions. In fact, it is a defining feature of life that it occurs far from equilibrium[27]. By enabling fast observations of non-equilibrium dynamics, microsecond time-resolved cryo-EM thus promises to fundamentally advance our understanding of living systems.

## Reporting summary

Further information on research design is available in the Nature Portfolio Reporting Summary linked to this article.

## Data availability

The data that support the findings of this study are available from the corresponding author upon request. The cryo-EM maps have been deposited in the Electron Microscopy Data Bank (EMDB) under accession codes EMD-16790 (extended CCMV), EMD-16798 (partially contracted CCMV), EMD-16400 (contracted CCMV); EMD-16857, (extended CCMV in UV irradiated sample, conventional), EMD-16858 (extended CCMV in UV irradiated sample, revitrified). The corresponding data set are accessible on EMPIAR (EMPIAR-11487, EMPIAR-11473, EMPIAR-11461, EMPIAR-11489, EMPIAR-11488). The atomic coordinates of the models have been deposited in the Protein Data Bank (PDB) under accession codes 8CPY (Extended CCMV) and 8C38 (contracted CCMV).

## Code availability

Computer code used to generate the results of this paper are available from the corresponding author upon request.

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

## Acknowledgements

We acknowledge the support of the Dubochet Center for Imaging (DCI) in Lausanne, as well as the Protein Production and Structure Core Facility at EPFL. We would also like to thank Prof. Jeroen Cornelissen and Dr. Regine van der Hee from the University of Twente for providing samples of CCMV, as well as Dr. Jonathan M. Voss and Dr. Pavel K. Olshin for help with the experiments, and Nathan J. Mowry for performing the simulations. This work was supported by the ERC Starting Grant 759145 and by the Swiss National Science Foundation Grant PP00P2_163681, both awarded to U.J.L.

## Author contributions

U.J.L. was responsible for conceptualizing this work. The methodology was performed by O.F.H., S.V.B., M.D., and U.J.L. O.F.H. performed cryo-

EM sample preparation and data acquisition. S.V.B. and O.F.H. performed the cryo-EM data processing. S.V.B performed the structure modeling and refinement. S.V.B., M.D, and U.J.L. performed the data analysis. Acquiring funding, project administration and supervision was performed by U.J.L. The writing of the original draft and data visualization was performed by S.V.B, O.F.H., and U.J.L. The reviewing and editing of the manuscript were performed by S.V.B, M.D., and U.J.L.

## Competing interests

The authors declare no competing interests.
