## [Peer Review File · Nature Communications]

Fast viral dynamics revealed by Microsecond Time-Resolved Cryo-EMReviewers' Comments:

Reviewer #1:

Remarks to the Author:

Structural analysis of protein dynamics is one of the main challenges in biology. All existing methodologies have limitations which severely hamper their usability for a wide range of biological systems. When assessing large structural re-arrangement which are usually not accessible in crystal systems, there is a paucity of methods and NMR or current methods in cryo-EM usually fail to address these processes.

The group of Ulrich Lorenz has developed in the last years a fast laser-based thaw-freeze method that allows to rapidly thaw a sample on the grid, trigger protein activation and re-vitrify the sample in microsecond time scales. In this paper they demonstrate for the first time that their method is appropriate to address biological questions which are not amenable to other time resolved methods.

Using their method the authors demonstrate vigorously that they can trap an intermediate state of the conformational change the viral capsid is undergoing during pH change. This intermediate state was so far not accessible to structural investigation. This is relevant to better understand the molecular mechanism of RNA release of CCMV during infection. The data is convincing and appropriate controls have been performed and included in the supplementary figures. This important part of proving the method is very well executed.

I strongly support publication of this work.

Detailed assessment and suggestions:

Some text modifications towards a stronger emphasis on the potential impact of the method and how it will be critical to address dynamics of other biological systems, which are not accessible at the moment, would help the reader to better appreciate the impact and high importance of the work.

A limitation of the current work is the limited resolution of the intermediate structure. The authors claim that this due to the conformational heterogeneity of the capsid proteins within viral particles and they are performing variability analysis to support their claim. When looking at the raw data in supplementary figure 1, the viewer notices that there is a lot of debris in panel b, which corresponds to the light activated, partially contracted conformation. This "debris" looks like it consists of broken up viral particles consisting of unspecified heterogenous oligomers. As the the quality of the control samples in supplementary fig. 2 look identical, the "debris" in supplementary figure 1 is most likely not related to the freeze-thaw cycle. It would be helpful if the authors discuss this issue.

One possibility might be that protonation of the negatively charged residues happen before any conformational change happen -as suggested by the authors- due to the protein being embedded in a vitrified sample. That may cause the built up of substantial internal tensions in the viral capsids which may cause disassembly rather than

conformational change upon rapid thawing in some particle. As the authors point out the expanded version of the virus is intrinsically less stable than the contracted one and the data might suggest that the sudden pH drop (which is inverse of the physiological event during infection) is further destabilizing the virus.

Overall, I am convinced that this paper presents good evidence that laser melting is a novel method to study changes in complex biological systems. I am also impressed with the simplicity of enabling this methodology in Cryo-EM workflows. I therefore expect that the method will be picked up by the Cryo EM structural biology community.

Reviewer #2:

Remarks to the Author:

Critique: This manuscript reports application of a novel method for preparing specimens for single particle cryoEM that achieves microsecond time resolution of dynamic processes. The authors have chosen their sample wisely using the pH dependent structural change in the capsid of Cowpea chlorotic mottle virus. Although most reactants will diffuse quite quickly in the volumes of a cryoEM specimen, protons are without a doubt, the fastest species to diffuse within an aqueous medium, making it extremely difficult to argue that the time resolution is hindered by species diffusion. By using an icosahedral virus, they facilitate high resolution 3-D imaging with a limited number of virus images.

I think that from the standpoint of technique development, this paper is highly significant. The technique as it has been reported in prior reports, really needed a good structural demonstration that it could achieve microsecond time resolution. I believe they have made that demonstration.

I only have minor comments which are made below which believe will improve the readability of the report.

Minor points:

I recommend that the authors deposit their raw data in EMPIAR in order that others who might be skeptical of the results be able to analyze them for themselves.

The paper would have been easier to review if the authors had included line numbers!!!!

Within the Supplemental Information:

Section A.

There are a few things that I must assume from the methods and which I would like clarified sufficiently to confirm that my assumptions are correct. (1) The freezing and thawing were not done under vacuum, but presumably in a water vapor-free environment. (2) If not, how long does the thawing and refreezing take and how much water can be expected to evaporate during this process? This is probably in one of the previous papers, but I would like this process to be explained more completely to make it easier on the reader to appreciate what has been accomplished here. They have plenty of space in the SI in which to do this.

Section B.

To spare the reader extra work, and to help maintain their surprise as to how this can be done, I suggest that you be more expansive about the time frames of the melting and revitrification sparing

them having to look up your prior publications.

"... was performed using Patch CTF." I assume that this is a routine within CryoSPARC. Saying so would be helpful.

" ... (I symmetry) ..." I symmetry? Don't you mean no symmetry, i.e. E symmetry? No biological molecule has I symmetry, which means it has a center of symmetry, except possibly in projection. Possibly you mean icosahedral symmetry. I don't believe there is a single letter designation for icosahedral symmetry among point group symmetries. I suggest you reword to "icosahedral symmetry".

Since you are not pressed for space in the SI, I suggest you be more expansive in the figure legends and tell the reader what they should be seeing.

We would like to thank the reviewers for their very meticulous reviews and are attaching our responses below.

Reviewer #1

Structural analysis of protein dynamics is one of the main challenges in biology. All existing methodologies have limitations which severely hamper their usability for a wide range of biological systems. When assessing large structural re-arrangement which are usually not accessible in crystal systems, there is a paucity of methods and NMR or current methods in cryo-EM usually fail to address these processes.

The group of Ulrich Lorenz has developed in the last years a fast laser-based thaw-freeze method that allows to rapidly thaw a sample on the grid, trigger protein activation and re-vitrify the sample in microsecond time scales. In this paper they demonstrate for the first time that their method is appropriate to address biological questions which are not amenable to other time resolved methods.

Using their method the authors demonstrate vigorously that they can trap an intermediate state of the conformational change the viral capsid is undergoing during pH change. This intermediate state was so far not accessible to structural investigation. This is relevant to better understand the molecular mechanism of RNA release of CCMV during infection. The data is convincing and appropriate controls have been performed and included in the supplementary figures. This important part of proving the method is very well executed.

I strongly support publication of this work.

Detailed assessment and suggestions:

Some text modifications towards a stronger emphasis on the potential impact of the method and how it will be critical to address dynamics of other biological systems, which are not accessible at the moment, would help the reader to better appreciate the impact and high importance of the work.

We take it as a compliment that the reviewer suggests that we are underselling the potential impact of our work, which we have tried to highlight more clearly in the manuscript.

A limitation of the current work is the limited resolution of the intermediate structure. The authors claim that this is due to the conformational heterogeneity of the capsid proteins within viral particles and they are performing variability analysis to support their claim. When looking at the raw data in supplementary figure 1, the viewer notices that there is a lot of debris in panel b, which corresponds to the light activated, partially contracted conformation. This "debris" looks like it consists of broken up viral particles consisting of unspecified heterogeneous oligomers. As the quality of the control samples in supplementary fig. 2 look identical, the "debris" in supplementary figure 1 is most likely not related to the freeze-thaw cycle. It would be helpful if the authors discuss this issue.

One possibility might be that protonation of the negatively charged residues happen before any conformational change happen -as suggested by the authors- due to the protein being embedded in a vitrified sample. That may cause the built up of substantial internal tensions in the viral capsids which may cause disassembly rather than conformational change upon rapid thawing in some particle. As the authors point out the expanded version of the virus is intrinsically less stable than the contracted one and the data might suggest that the sudden pH drop (which is inverse of the physiological event during infection) is further destabilizing the virus.

Yes, since the extended state is unstable, we observe a significant fraction of partially disassembled particles. This fraction does not seem to noticeably change upon revitrification of the sample. We have added a discussion of this point to the caption of Supplementary Fig. 1.

Overall, I am convinced that this paper presents good evidence that laser melting is a novel method to study changes in complex biological systems. I am also impressed with the simplicity of enabling this methodology in Cryo-EM workflows. I therefore expect that the method will be picked up by the Cryo EM structural biology community.

Reviewer #2

Critique: This manuscript reports application of a novel method for preparing specimens for single particle cryoEM that achieves microsecond time resolution of dynamic processes. The authors have chosen their sample wisely using the pH dependent structural change in the capsid of Cowpea chlorotic mottle virus. Although most reactants will diffuse quite quickly in the volumes of a cryoEM specimen, protons are without a doubt, the fastest species to diffuse within an aqueous medium, making it extremely difficult to argue that the time resolution is hindered by species diffusion. By using an icosahedral virus, they facilitate high resolution 3-D imaging with a limited number of virus images.

I think that from the standpoint of technique development, this paper is highly significant. The technique as it has been reported in prior reports, really needed a good structural demonstration that it could achieve microsecond time resolution. I believe they have made that demonstration.

I only have minor comments which are made below which believe will improve the readability of the report.

Minor points:

I recommend that the authors deposit their raw data in EMPIAR in order that others who might be skeptical of the results be able to analyze them for themselves.

We have in fact deposited our data on EMPIAR. The access codes are provided in the Data availability statement.

The paper would have been easier to review if the authors had included line numbers!!!!

We apologize for this oversight.

Within the Supplemental Information:

Section A.

There are a few things that I must assume from the methods and which I would like clarified sufficiently to confirm that my assumptions are correct. (1) The freezing and thawing were not done under vacuum, but presumably in a water vapor-free environment. (2) If not, how long does the thawing and refreezing take and how much water can be expected to evaporate during this process? This is probably in one of the previous papers, but I would like this process to be explained more completely to make it easier on the reader to appreciate what has been accomplished here. They have plenty of space in the SI in which to do this.

We have clarified these points in SI as follows.

We estimate that both the melting and the revitrification process occur on a timescale of about 5-7 μ s, as determined from heat transfer simulations, which we previously found to describe the experiment well². Evaporation of the sample in the vacuum of the electron microscope reduces its thickness by several tens of nanometers, with the change in thickness depending on the exact sample thickness and plateau temperature reached.

Section B.

To spare the reader extra work, and to help maintain their surprise as to how this can be done, I suggest that you be more expansive about the time frames of the melting and revitrification sparing them having to look up your prior publications.

As indicated in the previous point, we have added an estimate of the timescale of the melting and the revitrification process to the SI.

"... was performed using Patch CTF." I assume that this is a routine within CryoSPARC. Saying so would be helpful.

We have clarified this point in the manuscript.

“ ... (I symmetry) ...” I symmetry? Don't you mean no symmetry, i.e. E symmetry? No biological molecule has I symmetry, which means it has a center of symmetry, except possibly in projection. Possibly you mean icosahedral symmetry. I don't believe there is a single letter designation for icosahedral symmetry among point group symmetries. I suggest you reword to “icosahedral symmetry”.

We indeed mean to refer to icosahedral symmetry, with the Schoenflies symbol “I”. In order to avoid confusion with the point group “I_h”, we have opted to write “Icosahedral (I)”.

Since you are not pressed for space in the SI, I suggest you be more expansive in the figure legends and tell the reader what they should be seeing.

We have expanded the figure captions in the SI in order to make the figures more easily accessible.